# Semantic Localization System for Robots at Large Indoor Environments Based on Environmental Stimuli

**DOI:** 10.3390/s20072116

**Published:** 2020-04-09

**Authors:** Fco-Javier Serrano, Vidal Moreno, Belén Curto, Raul Álves

**Affiliations:** Department of Computer Science and Automation, University of Salamanca, 37008 Salamanca, Spain; fjaviersr@usal.es (F.-J.S.); bcurto@usal.es (B.C.); ralves@usal.es (R.Á.)

**Keywords:** indoor localization, particle filters, GIS maps, search and rescue tasks

## Abstract

In this paper, we present a new procedure to solve the global localization of mobile robots called Environmental Stimulus Localization (ESL). We propose that the presence of common facts on the environment around the robot can be considered as stimuli for the procedure. The robust performance of our approach is supported by two concurrent particle filters. A primary particle filter estimates and tracks the robot position, while a secondary filter is fired by environmental stimuli, helps to reduce the influence of measurement errors and allows an earlier recovery from localization failures. We have successfully used this method in a 5000 m${}^{2}$ real indoor environment using as inputs the available environment information from a Geographical Information System (GIS) map, the robot’s odometry and the output of an algorithm for the perception of facts from the environment. We present a case study and the result of different tests, showing the performance of our method under the influence of errors in real applications.

## 1. Introduction

Robot navigation in indoor environments constitutes the main challenge of search and rescue activities at urban locations (Urban Search and Rescue (USAR)) or, less dramatically but in a more usual fashion, at domestic or surveillance tasks. Disaster evacuation missions at an office building (for example an epidemiologic accident, bacteriologic attack, hazardous material leaking, etc.) require that the navigation task specification manages symbolic information with semantic meaning, like “the victim is found in room 424”. In this way, recent advances in Building Information Modelling (BIM) and Geographical Information System (GIS) suggest that the robot can makes use of the advanced semantic and geometric information included in an intelligent map, instead of the 2D geometric information of a map (mostly obtained with SLAM (Simultaneous Localization And Mapping approaches) [1,2]. So, it is clear that it would be quite useful for the robot to use the information from teh GIS systems directly. It can be also stated that tasks like rescue require that the robot can boast the global localization in order to provide an answer to questions like “where is the victim?” or “how to reach him?”.

Moreover, in USAR tasks, time is an important factor and the use of a robot can be restricted by the time that robot would need to explore and build a map. Proposals like [3] try to reduce it using emergency maps to support the SLAM strategy. Mur-Artal et al. [4] propose a visual inertial ORB-SLAM (ORB denotes an Oriented fast and Rotated BRIEF fact detection procedure) that is able to close loops in real time and can reuse the map, but it needs to perform the mapping task.

Our work tries to avoid this situation drastically with an original global location approach that combines a dual particle filter (PF) and a semantic map with natural landmarks obtained through GIS and BIM (Building Information Modelling) technologies

In this context, several algorithms approach this problem using grid-based Markov filters [5] or multi-hypothesis Kalman filters [6], but the most widely studied and tested algorithms are based on particle filters [7]. More recently, the approach provided by [8] uses particle filter to perform the localization task from MEMS (MicroElectroMechanical Systems) and camera devices, but it requires loop-use and the availability of previous image-formatted maps.

These filters, like other global localization algorithms, use the movement model and external perceptions to determine the robot’s localization (position and orientation). Modeling all the movements of a mobile robot with accuracy is very difficult because they depend on uncontrollable factors (shocks, slippages, etc.). In these situations, as it happens in the kidnapped robot problem, the PF can fail because all particles would be in the wrong place. Several authors have proposed modifications to the Monte–Carlo localization algorithm (MCL) [9], trying to resolve these situations. It requires that the number of added particles is large enough to be likely to hit the actual robot location. If this method is used for global localization in a very large environment, the number of particles required may be too high, or the filter may take a long time to converge to the correct position.

An improvement of this method is the SRL (Sensor Resetting Localization) algorithm [10], based on the addition of particles according to the probability distribution provided by sensor readings. While this method can improve the convergence of particle filters and solve the kidnapped robot problem in a very efficient way, it can also cause unwanted side effects when dealing with persistent erroneous measurements, like false positives detecting beacons, or errors in the map, like a beacon placed in a wrong position.

This paper describes a new localization method called Environmental Stimuli Localization (ESL), based on a dual particle filter. It consists of two PF running in parallel using the same data input: a primary or main filter and a secondary filter. The primary PF estimates and tracks the robot position, while the secondary filter has a short lifetime, has its own shooting and resetting conditions based on environmental stimuli and allows an earlier recovery from localization failures. In Section 3.2 this method and its implications will be explained in detail. As we will show, our method is also robust against persistent errors in perceptions, errors in the map or unmodelled robot movements.

There exist several works [11,12,13] where the dual particle filter idea is included, and more concisely at [14] it is applied to location tracking. All of them are based on the management of different state definitions at each filter where there are couplings of the respective prediction models. The data exchange between the filters is done through the shared parameters, or variables, of the models. In the opposite, at our proposal, both filters, Long-Term (LT) and Short-Term (ST) ones, consider the same state, that is, the robot position and orientation, and the same observations, doors or natural facts. They will have different firing conditions and execution (or duration) times. The data exchange mechanism is also established in an asynchronous way determined when the convergence of the ST filter is suitable. In this way, this is a new approach, as far as we know.

Another key aspect of this paper is the selection of the observation model. External measurements required by global localization algorithms are usually obtained by means of laser range finders [15], wireless transceivers [14,16] or cameras using pattern-detection algorithms [17,18]. In most cases, the implementation of these solutions requires a previous work on the whole robot environment, which implies taking images, mapping from laser readings, or inserting custom markers.

The use of inherent marks in the environment avoids this stage of preprocessing. Indoor, these marks could be rooms, windows, doors, walls, etc., while outdoors monuments, buildings, traffic signs, etc. could be considered. These marks should be matched with those extracted from the map of the robot’s navigation environment. In this work, we will present the application of the proposed ESL method and the observation model for a real mobile robot with wheels that travels through an interior environment and how it is able to work fine in large surface buildings (our campus). Among the distinguishing facts in indoor, we decided to use doors as external landmarks. Their main advantages are that they are quite abundant, since doors can be found in all rooms, and they are easier to detect than other objects since most of them share many common features. The size, shape and color of doors is fairly standard, especially within the same building, they are usually on the ground or over a doorstep, they usually have a frame, etc. Other objects, such as windows, present larger variations in shape, location, colors and sizes. Halls and walls are elements that have been widely used for localization tasks, but their detection is often achieved by means of a laser and it is more difficult to achieve with cheaper sensors like cameras. To detect doors, we use a method developed at our research group [19] that works well at a rate higher than 20 Hz and no environmental intervention is required. By means of a large number of experiments, our procedure has been proven to be very useful for confidently detecting nearby doors. Test scenarios included doors with different morphological characteristics as size, color, texture, shape, double and single doors, etc., objects with similar characteristics to the doors to test false positive and different livings spaces, as narrow and wide corridors, hallways, hall, among others.

Semantic facts, as doors at a building, are specifically represented at BIM and GIS systems. Our global localization solution does not require a manual pre-processing phase and it makes use of semantics facts automatically extracted from a GIS map, e.g., the doors localization on a given floor of the building. Currently, public and private institutions have services that maintain up-to-date GIS maps of indoor and outdoor areas, which can be downloaded or consulted online. The spatial databases that store GIS maps allow, among other advantages [20], to make queries on a wide variety of spatial operations

For example, by means of an automatic procedure, an on-line query to the building’s PostGIS database, we can obtain the location of the doors that are at a given distance, taking as origin the actual robot location. Hence, since it is possible to take advantage of this information to feed localization algorithms and is also possible for robots to detect doors, windows or other elements of those maps, it is feasible to implement global localization algorithms for a whole organization in a short time, without having to preprocess the environment or create custom maps.

When PFs are considered for global localization in large environments, one of the problems is that the number of particles required can be too high to run the algorithm in real-time. Without a proper method to initialize the position of the particles, in order to effectively sample the correct position of the robot, many thousands of particles can be required even for relatively small environments, just a few hundreds of square meters. USAR evacuation mission against an accident or hazardous material attack is posed at higher education buildings like the building at the Science campus of the University of Salamanca where our research group is located. The whole set of buildings in our University has an extension of many thousands of square meters, hence the number of particles required, using known approaches, could prevent the development of real-time global localization solutions. We will show how our proposed algorithm is able to deal with this problem. It is able to provide correct results in real-time doing global localization in a real environment using a low number of particles, even less than 1000, for a surface of more than 5000 m${}^{2}$ that corresponds to the Science Faculty building of our University where the robotic system was deployed.

The paper is organized as follows. First, we consider the main problems related to particle filters when they are applied to mobile robots. This analysis will be useful to contextualize the improvements of our approach. In Section 3 we present our proposal that, as a main result, improves the success rate of global localization tasks when working in large environments with a reasonable particle number and with erroneous maps or measurements. To prove it, we will present the real scenario where our localization tests are performed and, based on it, we will show the results of different tests and a case of study that evaluates the effectiveness of our method.

## 2. Particle Filters Applied to Robot Localization in Real Large Contexts

In this section, common problems that usually appear doing global localization of mobile robots in real environments will be discussed.

Sometimes, sensors produce unexpected measurements, which are not related to the information provided by our maps. When creating a map from laser measurements, objects such as furniture or paper bins, will appear in the generated map. If someone moves these objects, measurements from certain places won’t be as expected and therefore, localization algorithms will perform worse. When using computer vision techniques, as in our case to detect doors, these are especially prone to generate erroneous measurements. Depending on factors such as light, decoration, obstacles or the perspective of the camera, it is possible that certain objects are incorrectly detected, resulting in false positives or false negatives.

Within each of these two types of errors, we can differentiate two subtypes: persistent errors and short term errors. By short-term errors we mean those that happen during short periods of time (a few seconds) because of very specific circumstances of lighting, perspective, etc. that confuse the detection algorithm. By persistent errors we mean those that are caused by the presence of objects very similar to those we want to detect. Such objects cause errors in the detection algorithm that remain until they leave the robot’s field of view.

The effect of these unexpected measures on the probability density will be a probability decrease at the real robot location and a probability increment at locations that maximize $p\left(z_{t}|x_{t}^{\left(k\right)}\right)$ (where $z_{t}$ denotes the observations and $x_{t}^{\left(k\right)}$ is referred as the state or robot localization). Once unexpected readings disappear, the filter tends to converge again to the correct position. However, we should consider that these filters only have a finite number of particles and that, due to the resampling phase, they coalesce in places where the probability is higher. If during the time that wrong measurements are being received, all particles in the correct place disappear, the filter will not be able to recover and will fail in its estimation of the robot localization.

If erroneous measurements appear at the initialization phase of a filter and it is based on the probability distribution provided by sensor readings, the consequences can be fatal. It may be that, from the beginning, there are no particles placed near the robot’s real location or that, in the early stages of the filter, when there are a lot of places to explore and very few particles in each one, all particles near the real robot position disappear due to its worse matching with the erroneous measurements.

Less severe cases in which particles disappear from the right place can be usually solved by increasing the number of particles, by slowing down the convergence of the filter or by running the resampling phase only in some iterations of the filter. However, these solutions are not always feasible.

Increasing substantially the number of particles is not possible when we have a limited computing power, as in embedded systems, or when the localization must be performed on a very large map. However, there are optimizations that can help us to mitigate this problem like the usage of performance improvements [21] or adaptive sample sets [22].

Slowing down the convergence of the filter is not acceptable if you want to have a reliable estimation as soon as possible. In some cases, when using global localization modules, we may want to obtain a fast estimation more than a reliable estimation which requires too much time. In real applications, if we have a relatively accurate estimation of the robot’s position, we can activate the planning and navigation algorithms, so that the robot can start moving. While moving, the localization algorithm can continue working and finding the most accurate position, if the previous one was a wrong one.

The omission of the resampling phase in some of the iterations of the filter has a dual effect [23]. On one hand, it can prevent the disappearance of the particles located in the correct area. On the other hand, especially in the first iterations, it may cause the number of particles to track each possible robot position to become too small to successfully track the robot movement. If we want to avoid this, we must increase the number of particles according to the time we omit the resampling and to the dispersion of the initial probability distribution of the robot state.

When these problems cannot be fixed with these three approaches, we must restart the particle filter or add particles regardless of the normal execution. One possible solution is to reset the filter when we realize all the particles of the filter are at the wrong places. This can be detected when the sum of the particle weights before normalization is below a threshold. However, when using low accuracy sensors in environments with a lot of marks, it is very difficult for this solution to work properly in practice. If we set the threshold too low, the time to detect the localization failure can be too long because of casual coincidences between real and detected marks. Otherwise, if the threshold is too high, it is possible to reset the filter because of sensor errors even when it contains the correct robot location. It is, therefore, necessary to be very conservative when using this solution.

The addition of particles in the filter regardless of the normal execution is done in methods like SRL or Mixture-MCL [9] to enhance the performance of particle filters, but it has severe problems when dealing with unexpected sensor measurements, errors in the map or unmodeled robot movements.

## 3. Proposed ESL Method for Global Localization

Our approach is inspired by the natural behavior of humans trying to determine their own location. At any moment, a person can have a reasonable certainty of being in a specific location but, when a significant fact is perceived in the environment that conflicts with his estimation (maybe a monument or anything familiar to the person), if it is important enough, it can create a doubt and the person can start evaluating a second guess. Both the current estimation and the second guess can be evaluated in parallel and, if the doubt becomes important enough based on the next perceptions, the person can change his mind about his estimated position.

The core components of our proposed ESL localization method is detailed in Figure 1 where the architecture of our approach is presented. We propose the usage of a double concurrent particle filter and a GIS map as the sources of information, the output of a door detector [19] and the odometry of the robot. However, ESL filter doesn’t have any dependency on those sources of information, so our method should be easily applicable using others.

In order to explain how the ESL filter works, we will start showing how its state is evaluated. This evaluation is a key element in defining what the final output of our localization method is, and also the interaction between the two particle filters that run concurrently. This interaction allows the ESL filter to react to stimuli provided by the environment and, at the same time, it dramatically reduces the negative impact of sensor misreading, unmodeled robot movements or map errors. It will also be demonstrated that the number of particles required to get successful results using ESL is highly reduced compared to other approaches, so it is suitable to be applied in huge real environments.

### 3.1. Evaluation of Particle Filters

In order to save computing resources, ESL uses a simple clustering algorithm to group individual particles. Starting with an empty set of clusters, it iterates over the whole set of particles. When a particle is closer than a threshold to the center of an existing cluster, the particle is added to this cluster, its probability is added to the probability of this cluster, and the cluster center is updated to stay in the weighted mean position of the particles it contains. Otherwise, a new cluster is created with the same center and probability as the particle. Resulting clusters can be observed in Figure 2.

Using these clusters as input, the convergence of the filter is estimated using the entropy $H_{t}$ that is defined as
(1)$$H_{t}=-\sum_{cluster\;C}{\hat{p}}_{t}^{C}log_{2}\left({\hat{p}}_{t}^{C}\right),$$
where ${\hat{p}}_{t}^{C}$ represents the probability of each cluster *C*. When $H_{t}$ is high, the filter is too dispersed and needs to continue working to provide a reliable output. When $H_{t}$ is low, the convergence of the filter is high, with few clusters accumulating most of the probability. This value is essential for two different purposes: managing the interaction between the LT and the ST particle filters and providing a reliable output.

### 3.2. Concurrent Dual Particle Filter

When we consider the information provided by the environment (in our case the doors in front of the robot at a short distance), due to similarities and symmetries that appear in real environments, it is not possible to determine the localization of the robot based on the first detected doors. In these cases, if the available number of particles is low and the accuracy of sensors is not high enough, it is probable that one PF converges towards a wrong position. For that reason, it is important to take advantage of significant information that can appear in front of the robot at any time. This event has been referred to as stimulus in our approach. In our case, that significant information can be an accumulation of doors in few meters or a distribution of doors that is unlikely on other points of the map.

Since significant information or stimulus about the environment can be detected by the robot at any moment, that might not be the initial state, we propose a second particle filter (Short-Term PF, or ST). The ST filter is started every time that the environment provides significant information about the robot position.

As can be seen in Figure 1, we propose the usage of two particle filters running concurrently. The LT filter represents the certainty of being on each location of the map, so that the localization of the cluster with the highest probability (${\hat{p}}_{t}^{C}$) in the LT filter is considered the output of our algorithm when $H_{t}$ is below a certain threshold. The ST filter is intended to react to extra information provided by robot environment, stimulius, and provide that second guess, in this case a probability distribution, to the LT filter when the number of external measurements is higher than a given threshold. This environmental stimulus will be referred as the firing event of the ST filter Equation 2, defined as follows
(2)$${\mathrm{Inf}}_{t}>{\mathrm{Inf}}^{T},$$
where ${\mathrm{Inf}}^{T}$ is the selected threshold and ${\mathrm{Inf}}_{t}$ is defined as
(3)$${\mathrm{Inf}}_{t}=\mathrm{card}\left\{{\hat{z}}_{t}^{\left(j\right)}\right\}.$$

In our case, external observations $z_{t}^{\left(j\right)}$ are the doors in front of the robot detected by the method proposed by Fernandez–Carames et al. [19]. So ${\mathrm{Inf}}_{t}$ is the number of detected doors. Just after the firing event the ST filter is initialized according to the probability distribution provided by sensors (Section 3.3.1).

In this sense, our approach ESL is related to the SRL or Mixture-MCL proposal. The main difference is that, in those approaches, new particles are immediately introduced in the filter and, as we have commented in the Section 1, that causes severe problems when particles are added based on wrong external measurements. In our proposal ESL, the new particles are evaluated in the ST filter, without affecting the stability of the LT filter and the output of our algorithm.

To evaluate the quality of the particles in the ST filter, we use the filter entropy $H_{t}$, defined in Equation (Equation 1). When that value is low enough, the particles in the ST filter participate in the resampling phase of the LT filter, taking a low percentage of the available probability. After that, the ST filter is reset, waiting for the next firing event based on environmental stimulus.

So, we propose the reset condition as
(4)$$H_{t}<H^{T},$$
where $H^{T}$ will measure the maturity of the ST filter. This value should be low enough to ensure that the filter has received enough external information to be able to converge, grouping most of its particles in few clusters with a high probability. In our case, the value $H^{T}$ was experimentally determined.

Thanks to this hybrid resampling phase on the LT filter, some particles of those high-probability clusters of the ST filter start being evaluated in the LT filter. This makes a big difference between ESL compared to SRL or Mixture MCL. In those methods, added particles are introduced in the filter just based on a single observation (which could be wrong) and are immediately evaluated with very related sensor readings, giving new particles a clear advantage over others and worsening the negative effects of measurement errors. With the ESL method, the new particles have been already evaluated in the ST filter during a period of time, and are the result of the convergence of the ST filter. When these new particles start participating in the LT filter, current sensor readings are, with a high probability, not so related to the ones used to initialize them due to the time and robot movement that has happened during the maturation of these particles in the ST filter. These two factors reduce the amount of noise introduced in the LT filter and protect it against particles generated based on wrong external measurements or errors in the map.

Of course, with the ESL method, it is still possible to introduce wrong particles generated by wrong external measurements during the initialization of the ST filter but, in that case, those wrong particles stay isolated in the ST filter without affecting the stability of the LT one. When they are finally sampled by the LT filter, the external measurements used to evaluate them are much more independent of the measurements that generated them, so it is very unlikely that the wrong particles still evaluate better with current measurements than correct ones in the LT filter.

So, when the LT filter is wrong and few correct ST filter particles are added, it is probable that they cause the LT filter converges to the correct localization. When the ST filter just adds wrong particles, it is highly unlikely that they compromise the stability of the LT filter and cause it to fail. As we will show in the next section, the ESL approach has been successfully implemented and tested in a large real environment with a short response time and a very high success rate.

### 3.3. Additional Improvements in Standard Phases of Particle Filters

In order to improve our proposal, we have also introduced additional optimizations in the standard phases of particle filters applied to robot localization. In the initialization phase, we reduce the number of particles required to get successful results using the probability distribution provided by sensors. Additionally, we skip the resampling phase in some iterations of the filter to reduce the adverse effect of wrong external measurements. We explain these improvements in detail in the next sections.

#### 3.3.1. Initialization Phase

In order to locate the robot in a short period of time, the filter must sample the correct position and that usually requires the use of a huge amount of particles to explore a big surface (in our case, over 5000 m${}^{2}$). As the number of particles required by a filter usually increases with map dimensions, the required computational power to run the filter in real-time also increases. When the available hardware can not meet these requirements, additional solutions are needed to achieve successful results with a reduced set of particles.

The most used improvement in this sense is to select the initial particle set based on the probability distribution generated from the sensor readings ($p\left(x_{t}|z_{t}\right)$). This improvement was proposed as part of the SRL algorithm, in which the particles are placed in locations that maximize ($p\left(x_{t}|z_{t}\right)$).

Since we use a GIS map as a source of information, we use it to initialize the filter according to the probability distribution generated by sensor readings. Our external landmarks are doors perceived by our robot. To take advantage of them, only at the initialization phase for both filters, we group all detected doors in pairs, its relative distance is calculated and a search for pairs of doors with the same distance between them in the GIS map is performed, in order to achieve successful results with a reduced set of particles. Thanks to the use of a GIS map (stored in a PostGIS database), we obtain these pairs with a simple spatial query (Figure 3), where distance is the distance between both doors detected by the robot, and error is the estimated error for the measurement. The first condition in the WHERE clause prevents the evaluation of each pair of doors twice (in a different order) while the second and the third conditions select pairs of doors according to their distance.

Once we have the candidate door pairs, the detection of the robot is inversely applied over each door pair to calculate from which points of the map those doors could be seen in that way. Applying this method, each door pair provides two candidate positions for the robot on the map. To finish the initialization, the particles of the filter are distributed among those places, following a Gaussian distribution around each point. This way, the required number of particles to get a successful result does not depend on the extension of the map, but on the possible candidate positions, which is limited by the number of doors in the map and is much lower than its surface.

Therefore, the required number of particles is significantly reduced thanks to an initialization based on the instantaneous probability distribution provided by sensors and our map. This significantly reduces the time to locate the robot globally.

#### 3.3.2. Resampling Phase

The resampling phase of particle filters is intended to explore more efficiently the places of the map with a higher probability to be the actual position of the robot. However, it can also remove all particles from areas that have a low probability, but not zero, to be the current robot location. If all particles close to the actual robot position disappear, the filter will fail unless further actions are taken. An improvement to mitigate this undesired effect is to omit the resampling phase in some iterations. When the resampling is omitted, we must propagate the weights to the next iteration (Equation (Equation 5)) to not lose the information acquired during the correction phase.
(5)$$w_{t}^{\left(i\right)}=w_{t-1}^{\left(i\right)}p\left(z_{t}|x_{t}^{\left(i\right)}\right).$$

This modification has already been proposed in [23] to prevent loss of diversity in the positions of the particles and is implemented in the localization algorithm from the Carnegie Mellon Robot Toolkit (CARMEN), providing good results. In our algorithm, this improvement has also proved to be helpful in mitigating the negative effect of short term errors in measurements.

## 4. Results and Discussion

For tests, the Morlaco robot, designed at the Robotics and Society Group of the University of Salamanca (GroUsal) was used. It was equipped with a Sick LMS 200 laser range finder and an inexpensive Webcam Pro 9000 monocular camera from (Logitech, Lausanne, Swidish) Both sensors were connected to a Lenovo Thinkpad Edge laptop with an Intel Core i3 U380 CPU with two 2.54 GHz cores and 2GB of RAM (Lenovo, Beijing, China), where CARMEN was running.

Our proposed USAR task was developed at an office building where a hazardous material leaking, epidemiologc accident or bacteriological attack occurred and the building structure was not damaged. In this scenario, the Morlaco robot developed typical Search and Rescue activity. As Morlaco has heat, sound and video sensors, it could wander in the building at a victim search task so, finally, he can answer questions like: Where is the victim? or “how to reach him?”.

The tests were developed on the second floor of the Faculty of Sciences of the University of Salamanca, which has an extension of about 5000 m${}^{2}$ and a significant number of doors. Figure 2 shows the experimental environment where the experiments were performed. It is a large environment in which other approaches fail due to the very large number of required particles.

As a working procedure, we have moved the robot around different locations of the floor, saving all robot perceptions and odometry data using the logger tool from CARMEN. We have also provided the initial position of the robot to CARMEN to let it successfully track its movement and have an accurate estimation of the actual position of the robot to compare it with the result of our algorithm. Later, using the tool that CARMEN provides to replay logs, we ran multiple tests changing different configuration values and comparing the results.

In order to ease the understanding of the proposed ESL method, we are going to show a case study in detail. Later, our method will be compared to an SRL filter working in the same circumstances. Finally, the number of particles required by our approach will be compared to the usage of a random initialization, and the results of 50 different tests of the ESL method in our working environment will be shown.

### 4.1. Case Study

Among all tests we made, we are going to focus on one of them. This case study is especially interesting because it shows the behavior of our concurrent particle filter in difficult circumstances. The performance of the filter can be observed in the video (http://gro.usal.es/Localization.mp4), where the estimated position of the Morlaco robot is shown at the same time as the images of the camera used for the detection of doors (stimuli).

The video shows the robustness of the method in situations where no facts are detected (the doors are marked as small black circles) or even false facts are detected (false positives and negatives) due to perception errors or inaccuracies in the map, so the behavior of the procedure is shown in a realistic fashion. In the case that there exists no door in the robot’s surroundings for a long time period, the ST will not be fired, so only the LT filter will still be running, that is, existing particles will be propagated using the odometry although uncertainty will increase. However, at the time if a stimuli, such as doors, appears, the ST filter will be fired and in a short period it will introduce some particles with high quality (small uncertainty) to LT filter, and this one will get a more satisfying situation with better particles. In this way, the global localization process will be clearly enhanced It would be useless to compare localization algorithms in an ideal use-case in which the map is perfect and external measurements are accurate because all methods should succeed. Instead, during our case study, inaccurate perceptions force our double filter to demonstrate that it can provide good results even in that hard scenario. The ST filter is reset several times according to our method (eight times can be observed in the video). Due to inaccuracies in external measurements, the ST filter succeeded in some of the iterations and failed in others, but it never caused problems to the LT filter, that kept stable once the correct position of the robot was found.

Figure 2 shows the state of our concurrent particle filter just after the first resampling phase involving both filters. The big circles with a transparent background and different sizes and colors are the clusters of particles of each filter (the red color for LT filter and the green color for ST filter). The radius of the circle is related to the sum of the probabilities of the particles in the cluster ($P_{C}$). The actual localization of the robot at that moment was at the main horizontal corridor, near the fourth office starting from the left side, in the middle of the screenshot. The actual position is displayed as a big blue circle and the estimated position is shown as a big yellow circle, both overlapping in Figure 2.

In Figure 4, we can see the temporal evolution of our solution. The upper graph corresponds to the accuracy of both particle filters (the probability next to the correct location). The red line is associated with the LT filter and the green line corresponds to the ST filter. The second plot corresponds to the entropy $H_{t}$ (Equation (Equation 1)) for both filters. The third graph represents the number of detected doors $Inf_{t}$ (Equation (Equation 3)). It can be observed that most of the time a small number of facts are detected (one or even zero), but several times the firing event is met. The last graph shows the total distance traveled by the robot, so we can observe the amount of time and distance needed to get a correct localization. Firing events and resetting conditions are tagged in Figure 4 with vertical lines. All of them are also labeled at the video to get a complete understanding of the filter behavior.

As it can be seen in the video and in the Figure 4, few meters are needed (about 8 m, t = 35 s) to detect the correct position with a normalized accuracy (Equation (Equation 7)) close to 1, as it will be demonstrated later with a large set of executions. It is also very relevant that the correct localization is successfully maintained with time even, in situations where no information is available or the ST filter is wrong.

As can be observed, the initialization of the LT filter (t = 4 s) puts some particles in the correct position at the beginning of our test. In the vicinity of t = 18 s (about 4 m), the first firing event ($Inf_{t}>1$) starts the ST filter. After 7.5 m (t = 54 s), the entropy of the ST filter becomes low enough to trigger the resetting condition. We have experimentally determined that a reasonable threshold is $H^{T}=3$. The moment of the first resetting condition is spatially represented in Figure 2.

When particles of the ST filter are sampled in the resampling phase of the LT one (we reserve a 20% of the probability for it), the entropy of the LT filter raises a bit and, very quickly, the LT filter converges to the correct position. After that (around 7 s after), the firing event is met, so the ST filter starts a new iteration.

In order to evaluate the robustness of the method, we define the plausibility of the filter as
(6)$$V=\sum_{cluster\;C}\left\{A_{C}\;N_{C}\right\},$$
where $A_{C}$ is normalized accuracy of the cluster (the inverse of the distance from the centre of the cluster to the correct position)
(7)$$A_{C}=\min\left(\frac{1}{abs\left(D_{C}+\epsilon \right)},1\right)$$
being $D_{C}$ the distance from the cluster center to robot position and ϵ a small number greater than 0 to avoid singularities; and where $N_{C}$ is normalized size of the cluster
(8)$$N_{C}=\frac{P_{C}}{P_{max}}$$
being
$$P_{max}=\max_{cluster\;C}\left\{P_{C}\right\}.$$

In Figure 5, each cluster is represented by its normalized size $N_{C}$ in the horizontal axis and the normalized accuracy $A_{C}$ in the vertical axis. Clusters located in the actual position of the robot are green and others are red. Different plots show the state of the clusters (lost or not) for both filters when the resetting condition is met and the LT filter takes particles from the ST one. Each row of plots corresponds to a different resetting condition (timestamps at the video). For each row, the first plot represents the state of the LT filter before the hybrid resampling, the second one the state of the ST filter, and the third one the state of the LT filter after the hybrid resampling.

As it can be observed at the first timestamp, the value of *V* for the filter is not too high. This situation is described also in Figure 4. At the same time, the ST filter has recovered from the environment enough information to place particles on the correct location (a green cluster, although without the majority of particle population).

The result of the first resetting process is shown on the upper-right plot of the Figure 5a, with the LT filter having incorporated new particle clusters that let it explore places in the map that otherwise wouldn’t be sampled anymore, and therefore helping it to recover from errors without compromising its stability.

In the next resetting condition (timestamp 2) the few correct particles added by the ST filter (at the first timestamp) have evolved with the following observations and have managed to put all particles of the LT filter in the correct localization. Although the ST filter added wrong particles to the LT filter on the second (more than 60% of wrong particles) and the third (almost all the wrong particles) resetting conditions, the LT filter discarded them before the next iteration. So, we can see how the addition of wrong particles from the ST filter does not cause any problem to the LT one. Mainly, because it is very hard for wrong particles on the ST filter, that was added based on wrong external measurements from the past, to compete with correct particles in the LT filter being evaluated by independent external measurements (obtained several meters far from the others). In these two situations, the entropy of the LT filter will increase slightly, but the filter estimation remains robust over time and discards the incorrect ST filter particles in a short time.

### 4.2. Comparison ESL with SRL

In order to prove the feasibility of our ESL approach, we have compared it with the performance of an SRL implementation in the same scenario. Figure 6 shows how our proposal correctly detects the position of the robot while SRL is unable in a moment of that test. The complete execution can be observed on a video (http://gro.usal.es/ComparisonWithSRL.mp4). The continuous addition of particles based on the probability distribution provided by sensors helps to SRL filter to recover from errors (like our ST filter) but also destabilizes it when there are perception errors or inaccuracies in the map. In consequence, SRL lose track of the correct position of the robot several times, while our ESL keeps it very accurately during the whole test. So, our method keeps the best advantage of SRL but avoids its main inconvenience.

### 4.3. Number of Particles

Based on a previous case study, we have made several tests using ESL with a different number of particles to know how many are needed to get successful results and how many to get repeatable results in the course of the tests.

During the initialization phase of the LT and ST filters, the number of candidates to be the current localization of the robot, based on the doors detected and the GIS map, is between 199 and 375 in our case study. For that reason, we started using 400 particles on each filter. With a lower number of particles, it wouldn’t be possible to sample all candidate localizations, not even with one particle. We ran 10 tests with 400, 800, 1200, 2500 and 5000 particles per filter, using in all cases the same odometry and perceived data as in our case study.

The obtained results can be found in Table 1. The repeatability is defined from executing 10 times each initialization. Very low indicates, in a qualitative way, that the clusters obtained are completely distinct from one execution to others and the results vary significantly so the localization is unbelievable. Low indicates that the clusters do not present such a big variation but the result has lower quality. The high value is assigned when main clusters appear at every execution and finally, at very high, no difference between clusters can be appreciated. From Table 1, it can be concluded that when 5000 particles are considered, the localization result has great confidence.

Based on the previous case study, we have also run several tests to compare ESL initialization with random initialization with a different number of particles (Table 2). We started with 5000 particles because, due to the extension of our environment (about 5000 m${}^{2}$), it would be very unlikely to sample the correct position with a lower number of particles in random initialization. Using 5000 particles, our 10 tests failed. Table 2 shows that by using 50,000 particles, we got a 20% of success. With 100,000 particles, our CPU was already at 100% usage, but with a 60% of correct results. Using 200,000 particles, the delay in data processing was clearly noticeable. Probably for that reason, the number of successful tests was reduced to 30%. Tests with 500,000 particles made clear that the hardware was unable to process the data fast enough and provided wrong results in all cases.

To get results at least 60–70% of success, we needed 400 particles per filter using our ESL initialization method, and 100,000 using a random initialization. So the ESL method reduced around 250 times the number of needed particles. Our method also made it possible to get successful and repeatable results in all executions in our environment, while it was impossible for our hardware using a random initialization.

### 4.4. Success Rate

In order to know the success rate of the ESL method, we made 50 different tests to our case of study in order to be able to repeat and analyze the possible failures in different circumstances. During the tests, the Morlaco robot was moved between different places in our environment. We have used 5000 particles since our hardware can still run the algorithm in real-time. We have stored the error in the estimated position by ESL algorithm at each moment. In order to take into account not only the position of the robot but also its estimated orientation, we have considered each 20 degrees of deviation as 1 m of error.

Figure 7a shows the graphic representing the error of all tests (in meters) during a displacement of 12 m. We can see that the as meters the robot travels, more tests providing successful results will appear. For a better readability, Figure 7b shows the number of tests that provide a successful result (less than 2 m of error) versus the distance traveled by the robot.

In just 4 m, in 50% of the tests the robot was already correctly localized. After 9 m, in 90% of the tests, our method was providing the correct position of the robot. After 12 m, 98% of the tests were successful.

Since only one test failed after 12 m, we investigated it to see the cause. When we repeated that test using the same data, we saw that the first initialization failed due to false positives in the door detection algorithm. Some meters later, the ST filter was able to start sampling the correct position and sent some correct particles to the LT filter, but due to symmetries in the environment and inaccurate door detections, another position took more probability. The problem continued during several meters in which the robot was unable to perceive any door, until the 12 m that we considered the end of the test. Allowing the test to continue, as soon as the robot started detecting new doors, the localization algorithm started providing the correct localization, in this case after a total movement of 18 m. So, even in cases like this one, in which there were very adverse circumstances, our method was able to finally provide the correct localization of the robot as soon as it perceives significant information from the environment.

We consider that the fact that the robot gets a global localization in 90 % of the execution tests when the robot displacements are less than 9 m can be seen as very satisfactory with a temporal perspective when the large workspace (5000 m${}^{2}$) is considered. Any human in the same scenario would have this sensation. On the other hand, the measurement accuracy is supported by the door detection precision that, although influenced by the observation angle, is around 1 cm of precision [19].

## 5. Conclusions

In this work, we have proposed a new method for indoor robot localization based on two concurrent particle filters, one of which reacts to additional information provided by the environment (stimuli). The localization can be related within a BIM approach so this approach can be quite useful at search and rescue missions or domestic task completion.

The proposed method has been applied to the localization of a real robot working in a large and complex environment. Intense experimental work has been carried out with realistic tests in which the robot follows different paths. In 90% of the tests, after 9 m of travel, our method provided the correct position of the robot, despite persistent errors in perceptions, errors in the map or unmodeled robot movements. There are several reasons that explain and justify the results obtained by applying this algorithm.

First, the importance of the filter initialization. If we want to use a reduced number of particles in a large map, we must rely on the sensor measurements to place our particles in the most probable locations. If these measurements are erroneous, the initialization will be erroneous too, and the filter will probably fail. The stimuli based reinitialization of our ST filter gives the LT filter more chances to find the correct location of the robot.

Second, the independence between initialization data of the ST filter and correction data in the LT filter after the hybrid resampling (usually far enough in time and distance) prevents that the extra particles coming from the ST filter destabilize the LT one. In methods like SRL or Mixture-MCL, the new particles are added directly to the filter and are corrected using the same data used to generate them or, at least, with a strong conditional dependence (because they start being evaluated immediately after the addition). This gives new particles an advantage over the others, worsening the negative effects of measurement errors. With our method, even if the added particles were initialized based on wrong measurements, they do not cause the LT filter to converge to new erroneous locations and, at the same time, ESL still has the advantages of SRL or Mixture-MCL: fast recovery from errors or unmodeled robot movements. The robustness of our ESL approach has been proved on a real large indoor environment.

As future work, we could take into consideration the use of several secondary (or ST) filters where the stimuli are fired from different information sources or different kinds of characteristics or facts, like doors, columns or any other facts that can be contained at a GIS map. In this way, if we have environments where information is repeated (poor information in fact), like a corridor with symmetrically distributed doors or with a few doors, then we could use the other stimuli that provide useful information in such a way the particles will have more quality. A problem can appear if several filters are fired and the priority that would be assigned to these ones.

## Figures and Tables

**Figure 1 sensors-20-02116-f001:**
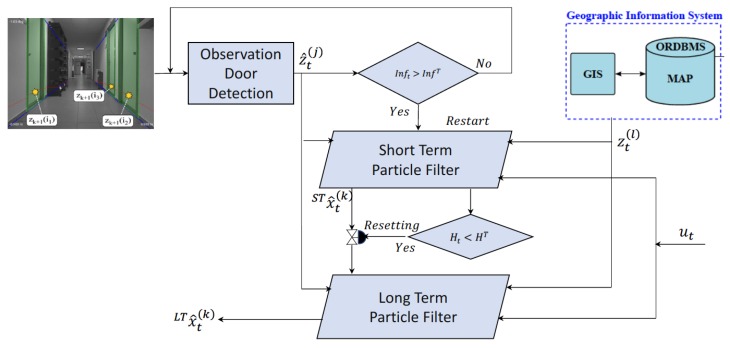
Core components of Environmental Stimuli Localization (ESL) method: the dual particle filter, Geographical Information System (GIS) map and landmark detector.

**Figure 2 sensors-20-02116-f002:**
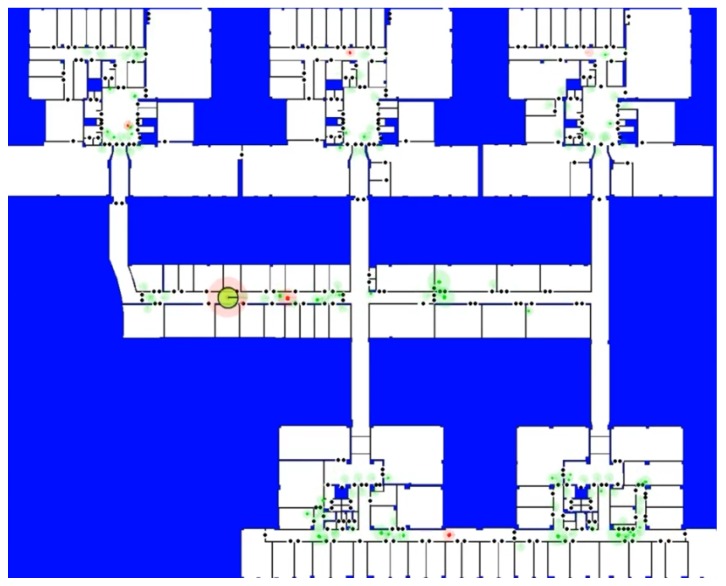
At the screenshot the building map where the different tests were developed. Black dots represent doors on the map. Circles represent the clusters and their size is related to their accumulated probability. Cluster color: green corresponds to the short-term (ST) filter and red represents the long-term (LT) filter. As it can be seen, the largest red cluster corresponds to final estimation of the robot position.

**Figure 3 sensors-20-02116-f003:**
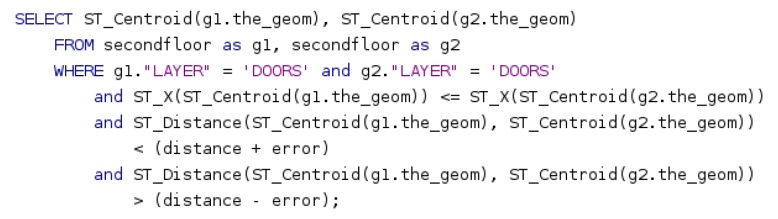
Query that is used to obtain the doors location at POSTGIS.

**Figure 4 sensors-20-02116-f004:**
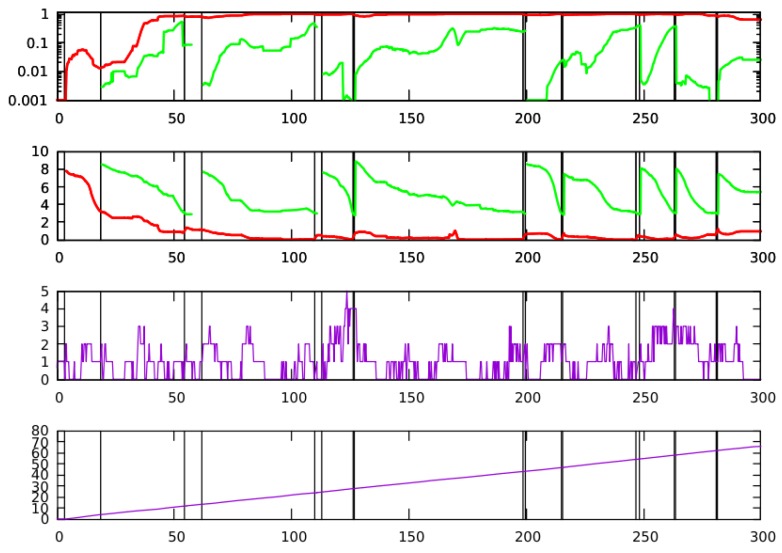
Time evolution (in seconds) of the concurrent filter. At first plot the normalized accuracy (red shows the LT filter and green shows the ST filter) is shown. Second plot represents the entropy (red shows the LT filter and green shows the ST filter). The third figure shows the number of detected doors. Fourth figure represents the travelled distance. Vertical lines represent the different times when the firing and resetting conditions are met.

**Figure 5 sensors-20-02116-f005:**
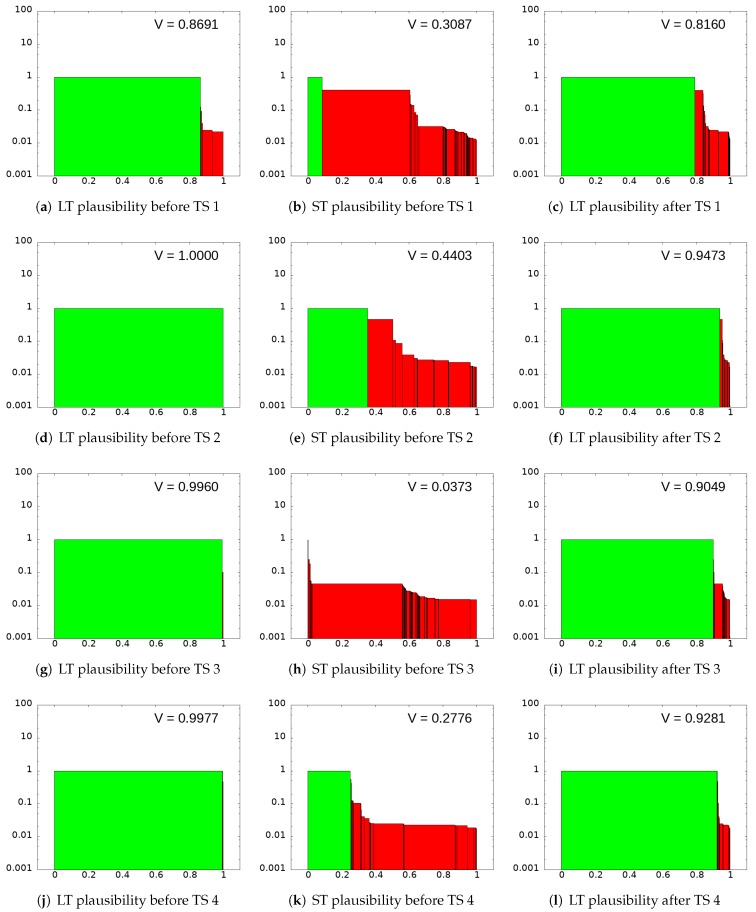
State of particle filters during the first four resetting conditions (timestamp or TS’s). It can be observed that the robustness of the procedure remains on time.

**Figure 6 sensors-20-02116-f006:**
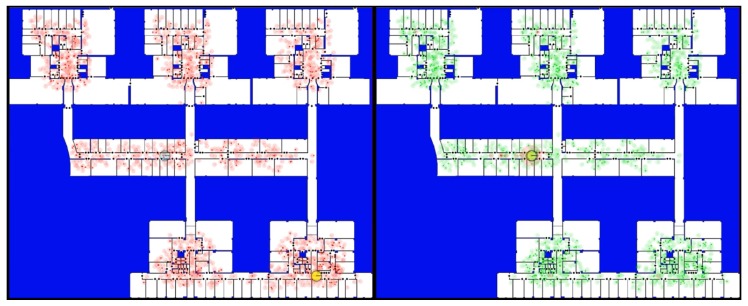
At the figure the localization based on SRL (left) is compared with our ESL filter (right). It can be seen that the robot on SRL is posed wrongly at a different position (bottom on the right) while on our ESL the robot is “strongly” placed (the red circle, in the middle, is large).

**Figure 7 sensors-20-02116-f007:**
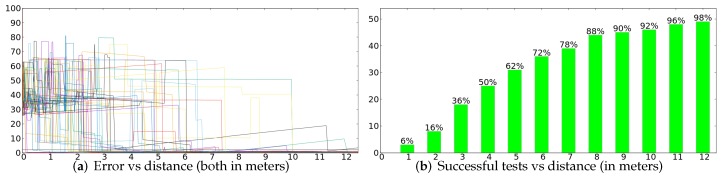
Results correspond to 50 executions where the robot has followed different paths

**Table 1 sensors-20-02116-t001:** Filter performance with our ESL initialization.

Particles	Success Rate	Repeatabilty
400	70 %	Low
800	90 %	Low
1200	100 %	High
2500	100 %	Very High
5000	100 %	Very High

**Table 2 sensors-20-02116-t002:** Filter performance with random initialization.

Particles	Success Rate	Repeatabilty
5000	0 %	Very Low
50,000	20 %	Very Low
100,000	60 %	Low
200,000	30 %	Very Low
500,000	0 %	Very Low

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
