# Peer review of "Semantic Localization System for Robots at Large Indoor Environments Based on Environmental Stimuli"

_sensors, 2020, doi:10.3390/s20072116_

Round 1

Reviewer 1 Report

The authors proposed a global localization system based on a dual particle filter paradigm, using automatically detected landmarks (doors) as part of the environmental information. The system is able to localize the robot after a relative small amount of traveled distance, in a moderately adverse environment that is quite large.

The manuscript is mostly clearly written, but has some issues that need to be acknowledged:

- The authors mention in line 28 that they are proposing an "original particle filter", and while the application of dual particle filters may be somewhat novel for localization purposes, it is not an "original" idea. There are works that have carried this idea in other areas, such as:

Hou, Dai-wen & Yin, Fu-liang. (2011). A Dual Particle Filter for State and Parameter Estimation in Nonlinear System. Journal of Electronics & Information Technology. 30. 2128-2133. 10.3724/SP.J.1146.2007.00273.

L.E. Olivier, B. Huang, I.K. Craig. Dual particle filters for state and parameter estimation with application to a run-of-mine ore mill. Journal of Process Control, Volume 22, Issue 4, 2012, Pages 710-717, ISSN 0959-1524.

Xingtao Liu, Zonghai Chen, Chenbin Zhang, Ji Wu. A novel temperature-compensated model for power Li-ion batteries with dual-particle-filter state of charge estimation. Applied Energy, Volume 123, 2014, Pages 263-272, ISSN 0306-2619.

Keunho Yun, Daijin Kim Robust location tracking using a dual layer particle filter. Pervasive and Mobile Computing, Volume 3, Issue 3, 2007, Pages 209-232, ISSN 1574-1192.

The authors should re-focus this statement to establish where their originality actually lies: the use of landmarks as part of the environment model that is used in the dual particle filter paradigm.

- The authors begin the manuscript mentioning applications such as search and rescue, but their case study centers around a large office environment. The authors should either add justification for their application in such an environment or establish how their case study is pertinent to a search and rescue scenario.

- The authors should state how are the true positions of the landmarks (which are assumed to be known a priori) to be established in the map. Is it a manual task? Or can there be an automatic procedure that could be carried out beforehand to avoid a manual input?

- Depending on architectural culture, it is not unusual in some countries to have different door sizes, colors and shapes in the same indoor environment. How can the proposed system be changed to use less specific criteria to define what a door is and where is located?

- The system uses pairs of doors (and the distance between them) as part of the information used by the second particle filter. Isn't this a very specific criterion of the environment? What happens in environments where doors are uniformly spaced?

- In line 351, the authors establish that an "accuracy close to 1" is acceptable. This needs to be justified.

- The numerical information shown in lines 403-423 should be presented in table form.

- In line 424, the authors establish that a 60-70% success rate is acceptable. This needs to be justified.

- Future work should be included in the conclusion section.

- There are many minor grammar and stylistic mistakes. It is recommended to submit the manuscript to a professional academic proofreading service to fix them. For example, here are the ones that this reviewer found in just the first two sections and the introduction of the third section:

. The popularly used term is "Search and Rescue", not "Rescue and Search".
. Figures are awkwardly placed. For example Figure 1 should be in section 3, Figure 2 should be in section 4, Figure 4 should be converted to text and not be a figure at all, Figure 7 is at the end of the manuscript (after references) and does not fit in the page.
. Videos should be provided as complementary material of the manuscript.
. line 13: "Robot navigation at indoor environment" -> "Robot navigation in indoor environments"
. line 27: "Our work try" -> "Our work tries"
. line 31: "the approach provided by [12] use" -> "the approach provided by [12] uses"
. line 32-33: "to reach the final information" -> do you mean "to obtain the final result", if so, it is redundant and should be removed
. line 121: don't use undefined variables to explain a concept. In this case, what is z_t, x_t and k? In this section it is recommended to just use a literal explanation, such as "and a probability increment at erroneous locations".
. line 129: "there no particles are placed" -> "there are no particles placed"
. line 152: "regardless the normal execution" -> "regardless of the normal execution"
. line 164: "is inspired on our idea that consider how the natural" -> "is inspired by the natural"
. line 171-172: "is detailed at Figure 1 where our architecture of approach is presented" -> "is detailed in Figure 1 where the architecture of our approach is presented"
. line 182: "to be applied to huge real environments" -> "to be applied in huge real environments"

Author Response

Thank you very much for your review comments. Please find our response at the attachment.​

Reviewer 2 Report

The paper discusses an interesting algorithm for the robot localization, based on two particle filters for the location estimation. Several concerns exist:

  1. Semantic localization is not a new concept. What is the actual motivation in terms of technical advances? 
  2. The paper has not discussed and compared performance with other SLAM-based algorithms? How does the proposed work achieve better performance? 
  3. It is not clear how the proposed system can run in real-time, given the target large indoor environments and so many particle filters. 

Author Response

(The authors gave the same response as above.)

Reviewer 3 Report

This paper concerns a solution of global localization of mobile robot, Environmental Stimuli Localization. It is worthwhile to do a detailed analysis on the state of the arts related to the localization. The authors presented the effectiveness of their proposed method via an experimental case study.

The following comments are to improve the presentation of the paper and to suggest the authors to emphasize the interests of their method.

[Minor modifications needed]

  1. It is necessary to place the picture numbers in the order in which they are mentioned in the text. For example, Figure 4 in Line 270 was mentioned before Figure 3 in Line 341.
  2. It is necessary to number the formulas given in Line 209 ~ 210.
  3. It is necessary to explain the variables in the equation in Line 121 and 292.
  4. The references used in the text should be listed in order.
  5. In line 350, figure has to be modified lowercase f to uppercase F.
  6. It is recommended that the websites in Line 321 and 393 be referred to the references.
  7. Too much () were used to explain further in the text as in line 15, 17, 19, 35, 37, 65 etc. It is necessary to minimize the use of () to help explain it in the main text.

[Major modifications needed]

  1. In line 66, the author insisted that doors be chosen as external landmarks for the proposed ESL method. If door does not exist for a quite a long time in an indoor environment, it is necessary to explain if your proposed method is still valid.
  2. The author insisted that the proposed global localization method is able to locate the robot accurately in a short time even with a small amount of particles. In line 441, they suggested that the robot was correctly positioned in about 90% of the test results after the robot moved 9 meters. It is necessary to explain why the results of locating the robot are claimed to be quick positioning methods. And it would be required to present in the text how much position measurement accuracy of the robot can be obtained by the proposed localization method.

Author Response

(The authors gave the same response as above.)
